# QWO: Speeding Up Permutation-Based Causal Discovery in LiGAMs

**Mohammad Shahverdikondori**

College of Management of Technology

EPFL, Lausanne, Switzerland

`mohammad.shahverdikondori@epfl.ch`

**Ehsan Mokhtarian**

School of Computer and Communication Sciences

EPFL, Lausanne, Switzerland

`ehsan.mokhtarian@epfl.ch`

**Negar Kiyavash**

College of Management of Technology

EPFL, Lausanne, Switzerland

`negar.kiyavash@epfl.ch`

## Abstract

Causal discovery is essential for understanding relationships among variables of interest in many scientific domains. In this paper, we focus on permutation-based methods for learning causal graphs in Linear Gaussian Acyclic Models (LiGAMs), where the permutation encodes a causal ordering of the variables. Existing methods in this setting do not scale due to their high computational complexity. These methods are comprised of two main components: (i) constructing a specific DAG, $\mathcal{G}^\pi$, for a given permutation $\pi$, which represents the best structure that can be learned from the available data while adhering to $\pi$, and (ii) searching over the space of permutations $\pi$s (i.e., causal orders) to minimize the number of edges in $\mathcal{G}^\pi$. We introduce QW-Orthogonality (QWO), a novel approach that significantly enhances the efficiency of computing $\mathcal{G}^\pi$ for a given permutation $\pi$. QWO has a speed-up of $O(n^2)$ ($n$ is the number of variables) compared to the state-of-the-art BIC-based method, making it highly scalable. We show that our method is theoretically sound and can be integrated into existing search strategies such as GRaSP and hill-climbing-based methods to improve their performance. The implementation is publicly available at `https://github.com/ban-epfl/QWO`.

## 1 Introduction

Causal discovery is fundamental to understanding and modeling the relationships between variables in various scientific domains [Pea09, SGSH00]. A causal graph, typically represented by a directed acyclic graph (DAG), is a graphical model that represents how variables within a system influence one another [Pea09]. The problem of causal discovery refers to learning the causal graph from available data [SGSH00, MT99, Chi02, FZ13, ZARX18, BSSU20a, MAGK21, LAR22, MAJ+22, MEAK24], which has broad applications across numerous fields such as economics [Hec08], genetics [SMD+03], and social sciences [MW15].

Score-based methods are an important class of approaches for causal discovery [JS10, KGG+22, HS13, HZL+18, Sch78, Bun91]. They involve defining a score function over the space of DAGs and seeking the graph that maximizes this score. However, the computational complexity of exploring the space of all possible DAGs, which is in the order of $2^{\Omega(n^2)}$, where $n$ is the number of variables, poses a significant challenge [Rob73]. Ordering-based methods, introduced by [TK05], significantly reduce

38th Conference on Neural Information Processing Systems (NeurIPS 2024).

Table 1: Time complexity comparison of various methods for computing and updating $\mathcal{G}^\pi$ in LiG-AMs. Here, $n$ is the number of variables, $d$ is the length of the updated block of the permutation, and $k$ is the number of folds considered in $k$-fold cross-validation.

| Method | QWO | BIC | BDeu | CV General |
|---|---|---|---|---|
| Initial Complexity | $O(n^3)$ | $O(n^5)$ | $\Omega(n^3 N \log(N))$ | $O(\frac{n^2 N^3}{k^2})$ |
| Update Complexity | $O(n^2 d)$ | $O(n^4 d)$ | $\Omega(n^2 d N \log(N))$ | $O(\frac{n d N^3}{k^2})$ |

the search space of score-based methods to $2^{\mathcal{O}(n \log(n))}$ by considering only topological orderings of DAGs, rather than the DAGs themselves [FK03, ZNC20, MKEK23, SLOT23]. Building on this premise, various permutation-based methods have been proposed that further refine the search for causal structures [SdCCZ15, LAR22, SWU21]. These methods have two main ingredients:

1. **Constructing $\mathcal{G}^\pi$:** A module that for any given permutation $\pi$ constructs a specific DAG, denoted by $\mathcal{G}^\pi$. We will formally define $\mathcal{G}^\pi$ in Definition 3.1, but roughly speaking, this DAG represents the *best* structure that can be inferred from the data while conforming to the given permutation $\pi$.

2. **Search over $\pi$:** The work by [RU18] (which proposed the Sparsest Permutation algorithm) showed that the correct permutation minimizes the number of edges in $\mathcal{G}^\pi$. Accordingly, permutation-based methods are equipped with a search strategy that, using the module mentioned above for computing $\mathcal{G}^\pi$, searches through the space of permutations to minimize the number of edges in $\mathcal{G}^\pi$.

In recent years, various search methods have been developed to enhance the accuracy, robustness, and scalability of the algorithm for the second part, i.e., searching over the permutations [LAR22, TBA06, MKEK23, TK05]. These methods typically involve traversing over the space of permutations by iteratively updating a permutation to reduce the number of edges in $\mathcal{G}^\pi$, using the first module to construct $\mathcal{G}^\pi$ multiple times during the algorithm's execution. To construct $\mathcal{G}^\pi$ or update $\mathcal{G}^\pi$ after a permutation update, most methods utilize a decomposable score function and identify the parent set that maximizes the score of each variable. Table 1 presents a time complexity comparison of different methods for computing and updating $\mathcal{G}^\pi$. We will delve into a more detailed discussion of these methods and their computational complexity in Section 3.

In this paper, we focus on linear Gaussian acyclic models (LiGAMs), an important class of continuous causal models. We propose a novel method called QW-Orthogonality (QWO) for computing $\mathcal{G}^\pi$ for a given permutation $\pi$, which significantly enhances the computational efficiency of this module. Specifically, as shown in Table 1, QWO's complexity is independent of the number of data points $N$. Moreover, it speeds up computing/updating $\mathcal{G}^\pi$ by $O(n^2)$ compared to the state-of-the-art BIC-based method. Some key advantages of QWO are as follows:

1. **Soundness:** QWO is theoretically sound for LiGAM models. That is, it is guaranteed to learn $\mathcal{G}^\pi$ accurately for any given permutation $\pi$ when sufficient samples are available.

2. **Scalability:** The proposed method is scalable to large graphs. Its time complexity is independent of the number of samples, and its dependence on the number of variables $n$ is $O(n^2)$ faster than the state-of-the-art BIC-based method.

3. **Compatibility with existing search methods:** After updating a permutation, QWO efficiently updates $\mathcal{G}^\pi$ to improve its compatibility with established search methods. In our experiments, we combine QWO with both GRaSP [LAR22] and a hill-climbing-based search technique [TBA06, SG06], showing its superior performance in terms of time complexity.

## 2   Notations

Throughout this paper, matrices are denoted by capital letters, and sets or vectors of random variables are denoted by bold capital letters. We use the terms "variable" and "vertex" interchangeably in graphs. $[n]$ denotes the set of natural numbers from 1 to $n$, and $\Pi([n])$ denotes the set of all $n!$ permutations over $[n]$. For a permutation $\pi \in [n]$, $P_\pi$ denotes the permutation matrix, such that for any $n \times n$ matrix $A$, the product $P_\pi A P_\pi^T$ permutes the rows and columns of $A$ according to the permutation $\pi$. The identity matrix is denoted by $I$, and its dimension is implied by the context.

$\|A\|_0$ denotes the number of non-zero entries in matrix $A$. The set of $n \times n$ diagonal matrices is represented by $\mathcal{D}_n$. $\langle a, b \rangle$ denotes the inner product of vectors $a$ and $b$. For a set $\mathbf{X} = \{X_1, \dots, X_n\}$ and $\mathbf{S} \subseteq [n]$, we define $\mathbf{X_S} = \{X_i | i \in \mathbf{S}\}$. diag$(A)$ denotes a diagonal matrix with the same diagonal elements as matrix $A$.

In a directed graph (DG), edges are directed and may form cycles. A DG without cycles is called a directed acyclic graph (DAG). If there is a directed edge from $X$ to $Y$, then $X$ is a parent of $Y$. $X$ is an ancestor of $Y$ if there is a directed path from $X$ to $Y$. In a DAG $\mathcal{G}$, for any vertex $X$, $e(\mathcal{G})$, $Pa_\mathcal{G}(X)$, and $Anc_\mathcal{G}(X)$ denote the number of edges, the parent set of $X$, and the ancestor set of $X$, respectively. Suppose $\pi$ is a permutation over the vertices of a DAG $\mathcal{G} = (\mathbf{X}, \mathbf{E})$. $\pi$ is a topological order for $\mathcal{G}$, or equivalently $\mathcal{G}$ is compatible with $\pi$, if for any edge $X_{\pi(i)} \to X_{\pi(j)} \in \mathbf{E}$, $i < j$.

**Definition 2.1** (LiGM, LiGAM, $G(B)$). *Suppose $\mathbf{X} = [X_1, \dots, X_n]^T$ is a random vector, $B$ is an $n \times n$ matrix such that $I - B$ is invertible, and $\Sigma \in \mathcal{D}_n$. Pair $\mathcal{M} = (B, \Sigma)$ is called a linear Gaussian model (LiGM) that generates $\mathbf{X}$ if*

$$\mathbf{X} = (I - B)^{-1}\mathbf{N}, \tag{1}$$

*where $\mathbf{N} \sim \mathcal{N}(0, \Sigma)$ is a Gaussian random vector with covariance matrix $\Sigma$. Equivalently, we have*

$$\mathbf{X} = B\mathbf{X} + \mathbf{N}. \tag{2}$$

*The causal graph of $\mathcal{M}$, denoted by $G(B)$, is a DG with the adjacency matrix corresponding to the support of $B^T$. A LiGM is a linear Gaussian acyclic model (LiGAM) if its causal graph is a DAG.*

The covariance matrix of $\mathbf{X}$ in a LiGM $\mathcal{M} = (B, \Sigma)$ is given by

$$\text{Cov}(\mathbf{X}) = (I - B)^{-1}\Sigma(I - B)^{-T}. \tag{3}$$

For distinct indices $1 \leq i, j \leq n$, and $\mathbf{S} \subseteq \mathbf{X}$, we use $X_i \perp\!\!\!\perp X_j | X_\mathbf{S}$ to denote that $X_i$ and $X_j$ are independent conditioned on $X_\mathbf{S}$. The notion of *d-separation* defined over DAGs is a graphical criterion to encode conditional independence (CI) within a graph. We similarly use $X_i \perp\!\!\!\perp X_j | X_\mathbf{S}$ to denote that $X_i$ and $X_j$ are d-separated given $X_\mathbf{S}$ in a DAG. For a formal definition of d-separation, see [Pea88].

**Definition 2.2** ([$\mathcal{G}$]). *Two DAGs are Markov equivalent if they impose the same d-separations. For a DAG $\mathcal{G}$, the set of its Markov equivalent DAGs are represented by [$\mathcal{G}$].*

In a LiGAM, the *Markov property* states that if two variables are d-separated in the DAG, then they are conditionally independent in the corresponding probability distribution. This property holds for structural equation models (SEMs), including LiGAMs; see Theorem 1.2.5 in [Pea09] for more details. Conversely, the *faithfulness* assumption posits that if two variables are conditionally independent in the distribution, then they are d-separated in the DAG. Together, the Markov property and faithfulness establish a one-to-one correspondence between the graphical d-separations and the conditional independencies in the distribution.

While the faithfulness assumption provides a strong correspondence between the distribution and the graph, it can be restrictive in practical applications. Recognizing this limitation, weaker versions of faithfulness have been proposed. One such alternative is the *sparsest Markov representation (SMR)* assumption introduced by [RU18]. Formally, for a DAG $\mathcal{G}$ and a distribution $P$ defined over the vertices of $\mathcal{G}$, $(\mathcal{G}, P)$ satisfies the SMR assumption if $(\mathcal{G}, P)$ satisfies the Markov property, and for every DAG $\mathcal{G}'$ such that $(\mathcal{G}', P)$ also satisfies the Markov property and $\mathcal{G}' \notin [\mathcal{G}]$, it holds that $|\mathcal{G}'| > |\mathcal{G}|$. Here, $|\mathcal{G}|$ denotes the number of edges in $\mathcal{G}$.

## 3 Permutation-Based Causal Discovery in LiGAMs

In this section, we discuss permutation-based methods for causal discovery in LiGAMs. We begin by formalizing our assumptions, which will be referenced throughout the remainder of the paper.

**Assumption 1.** *Let $\mathcal{M}^* = (B^*, \Sigma^*)$ be a LiGAM that generates the random vector $\mathbf{X} = [X_1, X_2, \dots, X_n]^T$. Let $D$ be an $N \times n$ data matrix, where each row is an i.i.d. sample from $\mathbf{X}$. We assume that the pair $(\mathcal{G}^*, P^*)$ satisfies the (SMR) assumption, where $\mathcal{G}^* = G(B^*)$ is the causal graph of $\mathcal{M}^*$ and $P^*$ is the joint distribution of $\mathbf{X}$.*

Under Assumption 1, our goal is to learn the causal graph $\mathcal{G}^*$ from the observational data $D$. However, using mere observational data—even when the number of samples $N$ approaches infinity—$\mathcal{G}^*$ can only be identified up to its Markov equivalence class (MEC) [SGSH00, Pea09]. Therefore, the best we can aim for in the causal discovery of LiGAMs is to identify $[\mathcal{G}^*]$, a problem known to be computationally NP-hard [CHM04].

As briefly discussed in the introduction, score-based methods aim to minimize a score function in the space of DAGs, whereas permutation-based methods restrict this space to topological orderings of DAGs. Thus, they seek a permutation $\pi$ such that $\pi$ is a topological order of a DAG in $[\mathcal{G}^*]$.

**Definition 3.1** ($\mathcal{G}^\pi$). *Under Assumption 1, for any arbitrary permutation $\pi \in \Pi([n])$, we denote by $\mathcal{G}^\pi = (\mathbf{X}, \mathbf{E}^\pi)$ the unique DAG constructed from $P^*$ as follows: For distinct indices $1 \leq i, j \leq n$, there is an edge from $X_{\pi(i)}$ to $X_{\pi(j)}$ in $\mathbf{E}^\pi$ if and only if*

$$i < j \quad and \quad X_{\pi(i)} \not\perp\!\!\!\perp X_{\pi(j)} | X_{\{\pi(1), \pi(2), \ldots, \pi(j-1)\} \setminus \{\pi(i)\}}. \tag{4}$$

**Remark 3.1.** *For any $\pi$, $\mathcal{G}^\pi$ is compatible with $\pi$. Furthermore, if a DAG $\mathcal{G} \in [\mathcal{G}^*]$ is compatible with $\pi$, then $\mathcal{G}^\pi = \mathcal{G}$.*

[RU18] showed that a permutation $\pi$ minimizes the number of edges in DAG $\mathcal{G}^\pi$ if and only if $\pi$ is the topological order of a DAG in $[\mathcal{G}^*]$. Therefore, permutation-based methods typically formulate causal discovery as follows.

$$\underset{\pi \in \Pi([n])}{\arg \min} |\mathbf{E}^\pi| \tag{5}$$

Various algorithms have been proposed for solving (5) [LAR22, SWU21, TK05, FK03]. As discussed in the introduction, these methods include two components:

1. **Constructing $\mathcal{G}^\pi$:** A module that for a given permutation $\pi$ constructs $\mathcal{G}^\pi$.

2. **Search over $\pi$:** A search strategy over the space of permutations to solve (5).

Given that minimization in (5), search strategies typically traverse the space of permutations and greedily update the permutation. To maintain computational efficiency, these methods use a module to update $\mathcal{G}^\pi$ incrementally rather than recomputing it from scratch. Moreover, good search strategies are ideally consistent in permutation-based causal discovery. A consistent search method guarantees to solve (5) as long as it is equipped with a module that correctly computes $\mathcal{G}^\pi$. In Appendix A, we review two search strategies: GRaSP [LAR22] and hill-climbing-based methods [TBA06, SG06]. GRaSP is consistent, ensuring that it finds the correct permutation that minimizes the number of edges in $\mathcal{G}^\pi$. In contrast, hill-climbing methods do not provide consistency guarantees but are shown to be efficient in practice.

Existing methods for computing $\mathcal{G}^\pi$ are primarily score-based and involve the following optimization:

$$\underset{\text{DAG } \mathcal{G}}{\arg \max} \quad S(\mathcal{G}; D, \pi)$$
$$\text{s.t.} \quad \mathcal{G} \text{ is compatible with } \pi, \tag{6}$$

where $S$ is typically a decomposable score function, summing individual scores for each variable given its parents within the graph:

$$S(\mathcal{G}; D, \pi) = \sum_{i=1}^{n} S(X_i, Pa_{\mathcal{G}}(X_i); D, \pi). \tag{7}$$

To find the parent set of each variable in $\mathcal{G}^\pi$, the following optimization is performed for all $1 \leq i \leq n$:

$$Pa_{\mathcal{G}^\pi}(X_{\pi(i)}) = \underset{\mathbf{U} \subseteq \{X_{\pi(1)}, \ldots, X_{\pi(i-1)}\}}{\arg \max} S(X_{\pi(i)}, \mathbf{U}; D, \pi). \tag{8}$$

To solve (8), various approaches have been proposed. Note that the complexity of a brute-force search over all subsets $\mathbf{U}$ is exponential, which makes it impractical. Instead, the state-of-the-art search methods apply the *grow-shrink (GS)* algorithm [ARSR+23] on the candidate sets $\mathbf{U}$ to find the parent set of each variable. These methods require computing the score function $S$, $O(n^2)$ times.

It is noteworthy that the chosen score function must ensure that the solution to this optimization problem equals $\mathcal{G}^\pi$. Examples of such scores include BIC [Sch78], BDeu [Bun91], and CV General [HZL+18]. Table 1 compares the time complexity of our proposed method (QWO) against

these methods. The Bayesian Information Criterion (BIC) is a well-known score in the literature, particularly for LiGAMs. BIC's complexity for calculating the initial graph is $O(n^5)$; the update is $O(n^4 d)$. The Bayesian Dirichlet equivalent uniform (BDeu) score applies a uniform prior over the set of Bayesian networks and uses this prior to evaluating the model's accuracy. While BDeu was originally designed for discrete data, it has been adapted for continuous data by dividing the real numbers into intervals and assigning a constant value for each interval. The time complexity of BDeu depends on the number of distinct values each variable can take (i.e., intervals). Its initial and update complexities are lower-bounded by $\Omega(n^3 N log(N))$ and $\Omega(n^2 dN log(N))$, respectively. The Generalized Cross-Validated Likelihood (CV General) score involves splitting the dataset into training and test sets multiple times. The final score for a variable given its parents is the average log-likelihood evaluated on the test sets using the regression functions learned from the training data. The CV General method has an initial complexity of $O(\frac{n^2 N^3}{k^2})$ and an update complexity of $O(\frac{ndN^3}{k^2})$. This method typically uses a small constant $k$ for $k$-fold cross-validation, which significantly increases the computational complexity. Among the aforementioned three strategies, BIC is the fastest. Our proposed method, QWO, attains a speed-up of $O(n^2)$ compared to BIC.

# 4 QWO

In this section, we present a novel approach for computing $\mathcal{G}^\pi$ for a permutation $\pi$, with improved computational complexity, which can be easily integrated into existing search methods. Our proposed method proceeds with an alternative formulation for causal discovery in LiGAMs, but first, we need a definition.

**Definition 4.1** ($\mathcal{B}(\mathbf{X})$)**.** *For a random vector $\mathbf{X}$, we denote by $\mathcal{B}(\mathbf{X})$ the set of coefficient matrices of LiGMs that generate $\mathbf{X}$, i.e.,*

$$\mathcal{B}(\mathbf{X}) = \{B | \exists \Sigma \in \mathcal{D}_n : \quad Cov(\mathbf{X}) = (I - B)^{-1}\Sigma(I - B)^{-T}\}.$$

Note that in this definition, the causal graphs corresponding to the elements of $\mathcal{B}(\mathbf{X})$ can be cyclic, but we restrict our attention to a subset of $\mathcal{B}(\mathbf{X})$ with acyclic corresponding graphs. We reformulate causal discovery in LiGAMs as follows:

$$\begin{aligned} \underset{B \in \mathcal{B}(\mathbf{X})}{\arg \min} \quad & \|B\|_0. \\ \text{s.t.} \quad & G(B) \text{ is a DAG} \end{aligned} \tag{9}$$

It has been shown that for any solution $B$ of (9), $G(B)$ belongs to $[\mathcal{G}^*]$. Furthermore, for any graph $\mathcal{G} \in [\mathcal{G}^*]$, there exists a solution $B$ to (9) such that $G(B) = \mathcal{G}$ [Pea09]. Therefore, solving (9) is equivalent to performing causal discovery in LiGAMs.

In the following, we establish the relationship between $\mathcal{B}(\mathbf{X})$ and $\mathcal{G}^\pi$.

**Theorem 4.2.** *Under Assumption 1, for any permutation $\pi \in \Pi([n])$, there exists a unique $B \in \mathcal{B}(\mathbf{X})$ such that $G(B)$ is compatible with $\pi$. Furthermore, for this $B$, $G(B) = \mathcal{G}^\pi$.*

All proofs are provided in Appendix C. Theorem 4.2 implies that to compute $\mathcal{G}^\pi$, we can directly find the unique $B \in \mathcal{B}(\mathbf{X})$ whose corresponding graph $G(B)$ is compatible with $\pi$. Next, we propose a characterization for $\mathcal{B}(\mathbf{X})$ using whitening transformation [Fuk90, HO00, KLS18].

**Definition 4.3** (Whitening matrix $W$)**.** *Let $Cov(\mathbf{X}) = USU^T$ be the singular value decomposition (SVD) of $Cov(\mathbf{X})$, where $S$ is a diagonal matrix including the singular values of $Cov(\mathbf{X})$ on its diagonal and $U$ is an orthonormal matrix (i.e., $UU^T = I$). The whitening matrix $W$ is defined as*

$$W := US^{-\frac{1}{2}}U^T. \tag{10}$$

We note that the 'W' in QWO corresponds to the whitening matrix $W$. Whitening is a linear transformation $\mathbf{N}_I := W\mathbf{X}$ that transforms the Gaussian random vector $\mathbf{X}$ to another Gaussian random vector $\mathbf{N}_I$, where $Cov(\mathbf{N}_I) = I$. Furthermore, for an arbitrary orthogonal matrix $Q$ (i.e., $QQ^T \in \mathcal{D}_n$), if we define $\mathbf{N}_Q := Q\mathbf{N}_I = QW\mathbf{X}$, then it is straightforward to show that $\mathbf{N}_Q$ is also a Gaussian random vector with mean zero and a diagonal covariance matrix. Therefore, $(I - QW, \mathbf{N}_Q)$ is a LiGM that generates $\mathbf{X}$ since

$$\mathbf{X} = (I - QW)\mathbf{X} + \mathbf{N}_Q.$$

In the following, we show that such LiGMs create all possible LiGMs that generate $\mathbf{X}$.

---

**Algorithm 1** The QWO module for computing and updating $\mathcal{G}^\pi$

---

1: **Function QWO**($\pi$, $W$, [Optional] idx$_l$, [Optional] idx$_r$, [Optional] $Q$)
2: **Default values for optional arguments:** idx$_l = 1$, idx$_r = n$, $Q =$ any arbitrary $n \times n$ matrix
3: Denote the columns of $Q^T$ by $\{q_i\}_{i=1}^n$ and the columns of $W$ by $\{w_i\}_{i=1}^n$
4: **for** $i$ from idx$_r$ to idx$_l$ **do**
5:    $r_i \leftarrow w_{\pi(i)} - \sum_{j=i+1}^n \frac{\langle w_{\pi(i)}, q_{\pi(j)} \rangle}{\|q_{\pi(j)}\|_2^2} q_{\pi(j)}$
6:    $q_{\pi(i)} \leftarrow \frac{r_i}{\langle r_i, w_{\pi(i)} \rangle}$
7: **end for**
8: $\mathcal{G}^\pi \leftarrow G(I - QW)$
9: **Return** $\mathcal{G}^\pi, Q$

---

**Theorem 4.4** (Characterizing $\mathcal{B}(\mathbf{X})$). *For any $B \in \mathcal{B}(\mathbf{X})$, there exists a unique orthogonal matrix $Q$ such that $B = I - QW$, and vice versa. That is,*

$$\mathcal{B}(\mathbf{X}) = \{I - QW \,|\, QQ^T \in \mathcal{D}_n\}. \tag{11}$$

By combining Theorems 4.2 and 4.4, to learn $\mathcal{G}^\pi$, it is sufficient to identify the unique orthogonal matrix $Q$ such that $G(I - QW)$ is compatible with $\pi$. To verify this compatibility, we impose the following two constraints:

$$P_\pi QW P_\pi^T \text{ is upper triangular}, \quad \text{diag}(P_\pi QW P_\pi^T) = I. \tag{12}$$

## 4.1 The QWO Method

In this subsection, we introduce QW-Orthogonality (QWO), our proposed approach for computing $\mathcal{G}^\pi$ in LiGAMs. The function QWO in Algorithm 1 presents the pseudocode of our method. The function takes a permutation $\pi$ and a whitening matrix $W$ as inputs, with optional arguments idx$l$, idx$r$, and $Q$. First, we consider the case where these optional arguments are not provided, and they are initialized as follows: idx$_l = 1$, idx$_r = n$, and $Q$ to be an arbitrary $n \times n$ matrix. The goal of this function is to create a matrix $Q$ that is the unique solution of (12) and subsequently to compute $\mathcal{G}^\pi$.

Denote the columns of $Q^T$ by $\{q_i\}_{i=1}^n$ and the columns of $W$ by $\{w_i\}_{i=1}^n$. Since $W = US^{-\frac{1}{2}}U^T$, all the eigenvalues of $W$ are positive, and thus $\{w_i\}_{i=1}^n$ are $n$ linearly independent vectors in an $n$-dimensional space. Furthermore, because $Q$ is an orthogonal matrix for each pair $(i,j)$, $1 \leq i \neq j \leq n$, $q_i \perp q_j$. The condition $P_\pi QW P_\pi^T$ is upper triangular in (12) implies that $q_{\pi(i)}$ should be orthogonal to $w_{\pi(j)}$ if $j > i$. Therefore, we need to find $\{q_i\}_{i=1}^n$ such that this condition holds. Note that each zero in $\|I - QW\|_0$ corresponds to an orthogonality between $\{q_i\}_{i=1}^n$ and $\{w_i\}_{i=1}^n$.

With this intuition, we propose our approach for constructing $Q$, which maximizes the number of aforementioned orthogonality. To do so, we apply the Gram-Schmidt algorithm[GL13] to the vectors $\{w_i\}_{i=1}^n$ in the following order: $\pi(n), \pi(n-1), ..., \pi(1)$. In Algorithm 1, we iteratively for each $i$, compute $r_i$, the residual of projecting $w_{\pi(i)}$ on the span of $\{w_{\pi(i+1)}, ..., w_{\pi(n)}\}$. This is equivalent to projecting $w_{\pi(i)}$ on the span of $\{q_{\pi(i+1)}, ..., q_{\pi(n)}\}$, which are orthogonal to each other. $q_{\pi(i)}$ is set to normalized residual $r_i$, which ensures that the product of $q_{\pi(i)}$ and $w_{\pi(i)}$ is 1, and consequently, $\text{diag}(P_\pi QW P_\pi^T) = \text{diag}(QW) = I$. Finally, we use Theorem 4.2 to construct $\mathcal{G}^\pi = G(I - QW)$ and return $\mathcal{G}^\pi$ and $Q$.

**Theorem 4.5** (Soundness of Algorithm 1). *Under Assumption 1 and given the correct whitening matrix $W$ as input, matrix $Q$, the output of Algorithm 1 is the unique solution to* (12). *Consequently, the returned graph corresponds to the true $\mathcal{G}^\pi$ defined in Definition 3.1.*

The optional arguments idx$_l$, idx$_r$, and $Q$ allow for incremental efficient updates to $\pi$.

**Lemma 4.6.** *If the block between the idx$_l$-th and idx$_r$-th positions of $\pi$ is modified, the vectors $q_{\pi(k)}$ for $k < idx_l$ or $k > idx_r$ remain unchanged.*

A consequence of Lemma 4.6 is that $q_{\pi(k)}$ for $k < $ idx$_l$ or $k > $ idx$_r$ from the previous computed $Q$ can be reused and it sufficed to merely recompute the vectors within the updated block by iterating through the for loop in lines 4-7.

---

**Algorithm 2** Integrating QWO into a simple search method for causal discovery

---

1: **Input:** Data matrix $D$, Search method $\mathcal{F}$
2: Estimate $\text{Cov}(\mathbf{X})$ using data matrix $D$
3: $U, S \leftarrow$ Compute the SVD decomposition of $\text{Cov}(\mathbf{X})$, such that $\text{Cov}(\mathbf{X}) = USU^T$
4: $W \leftarrow US^{-\frac{1}{2}}U^T$                                                 % Whitening matrix
5: $\pi \leftarrow$ An initial permutation
6: $\mathcal{G}^\pi, Q \leftarrow \mathbf{QWO}(\pi, W)$
7: **while** $\mathcal{F}$ has not stopped **do**
8:     $\pi' \leftarrow$ Update $\pi$ according to $\mathcal{F}$, and let $\text{idx}_l$ and $\text{idx}_r$ be the leftmost and rightmost index of $\pi$ that have been updated, respectively
9:     $\mathcal{G}^{\pi'}, Q' \leftarrow \mathbf{QWO}(\pi, W, \text{idx}_l, \text{idx}_r, Q)$
10:     **if** $\mathcal{G}^{\pi'}$ has less number of edges than $\mathcal{G}^\pi$ **then**
11:         $\pi, Q, \mathcal{G}^\pi \leftarrow \pi', Q', \mathcal{G}^{\pi'}$
12:     **end if**
13: **end while**
14: **Return** $\mathcal{G}^\pi$

---

**Theorem 4.7** (Time complexity of Algorithm 1). *QWO algorithm as implemented in Algorithm 1 has the following time complexities:*

- $O(n^3)$ *for initially computing* $\mathcal{G}^\pi$ *without optional arguments.*

- $O(n^2 d)$ *when called with optional arguments to update* $\mathcal{G}^\pi$*, where* $d = \text{idx}_r - \text{idx}_l$.

### 4.2 Integrating QWO in Existing Search Methods

Algorithm 2 demonstrates how QWO can be integrated into existing search methods such as GRaSP [LAR22] and Hill-Climbing [TBA06, SG06]. This algorithm integrates QWO into a simple permutation-based search method for causal discovery that iteratively updates the permutation to minimize the number of edges in $\mathcal{G}^\pi$. Initially, the algorithm estimates the covariance matrix of $\mathbf{X}$ and applies SVD to compute the whitening matrix. Starting from an initial permutation, it calls function QWO to compute $Q$ and $\mathcal{G}^\pi$. Subsequently, the algorithm iteratively updates the permutation using the given search method $\mathcal{F}$[1]. For each updated permutation, it calls function QWO with optional arguments to compute the new $Q$ and $\mathcal{G}^\pi$ graph. Next, the algorithm checks if the new permutation is better than the previous one by checking whether the new graph has fewer edges. If the new permutation is better, it updates the permutation and proceeds to the next iteration. The search algorithm stops when its stopping criterion is satisfied and returns the final $\pi$ and $\mathcal{G}^\pi$.

## 5 Experiments

In this section, we present a comprehensive evaluation of QWO, which is designed for the module that computes $\mathcal{G}^\pi$ in permutation-based causal discovery methods in LiGAMs. As discussed earlier, a permutation-based method consists of two components: a search method and a module for computing $\mathcal{G}^\pi$. Our comparison focuses on the module for computing $\mathcal{G}^\pi$, where we benchmark QWO against existing methods, namely BIC [Sch78], BDeu [Bun91], and CV General [HZL+18]. For the search method, we utilized two specific algorithms: GRaSP [LAR22] and Hill-Climbing (HC) [TBA06, SG06]. Please refer to Appendix B for the implementation details of these methods and additional results.

We generated random graphs according to an Erdos-Renyi model with an average degree of $i$ for each node, denoted by $ERi$. We did not impose any constraints on the maximum degree of the nodes. To generate the data matrix $D$ using a LiGAM $(B^*, \Sigma^*)$, we sampled the entries of $B^*$ from $[-2, -0.5] \cup [0.5, 2]$ and the noise variances uniformly from $[1, 2]$. Each reported number on the plots is an average of 30 random graphs.

---

[1]Search method $\mathcal{F}$ could be any existing approach such as GRaSP [LAR22] or Hill-Climbing [TBA06, SG06].

Table 2: Results for small graphs using GRaSP and Hill-Climbing as search methods.

| Graph | | | CANCER | SURVEY | ASIA | SACHS | ER2 |
|---|---|---|---|---|---|---|---|
| Number of Nodes | | | 5 | 6 | 8 | 11 | 5 |
| Average Degree | | | 1.6 | 2 | 2 | 3.09 | 2 |
| QWO | GRaSP | SKF1 | 1 | 1 | 0.87 | 0.88 | 0.85 |
| | | PSHD | 0 | 0 | 0.87 | 0.63 | 0.2 |
| | | Time (s) | 0.003 | 0.004 | 0.01 | 0.04 | 0.002 |
| | HC | SKF1 | 0.88 | 0.92 | 0.94 | 0.88 | 0.97 |
| | | PSHD | 0.6 | 0.5 | 0.625 | 0.63 | 0.4 |
| | | Time (s) | 0.01 | 0.01 | 0.01 | 0.05 | 0.003 |
| BIC | GRaSP | SKF1 | 1 | 1 | 1 | 0.93 | 1 |
| | | PSHD | 0 | 0 | 0 | 0.81 | 0 |
| | | Time (s) | 0.06 | 0.07 | 0.08 | 0.39 | 0.02 |
| | HC | SKF1 | 0.88 | 0.61 | 0.94 | 0.93 | 0.97 |
| | | PSHD | 0.8 | 1.33 | 0.625 | 0.54 | 0.4 |
| | | Time (s) | 0.01 | 0.05 | 0.08 | 0.83 | 0.15 |
| BDeu | GRaSP | SKF1 | 0.25 | 0.36 | 0.26 | 0.29 | 0.12 |
| | | PSHD | 1.4 | 1.5 | 1.5 | 1.72 | 1.4 |
| | | Time (s) | 44.1 | 72.1 | 116.6 | 211.6 | 41.1 |
| | HC | SKF1 | 0.5 | 0.54 | 0.4 | 0.37 | 0.6 |
| | | PSHD | 1.2 | 1.16 | 1.374 | 1.81 | 1.2 |
| | | Time (s) | 64.4 | 95.1 | 224.4 | 496.6 | 78.2 |
| CV General | GRaSP | SKF1 | 1 | 0.57 | 0.76 | 0.87 | 0.85 |
| | | PSHD | 0 | 1.66 | 1.12 | 0.90 | 0.2 |
| | | Time (s) | 109.6 | 296.3 | 641.1 | 867.5 | 101.6 |
| | HC | SKF1 | 1 | 0.5 | 0.88 | 0.87 | 0.9 |
| | | PSHD | 0 | 1.33 | 1.12 | 0.72 | 0.6 |
| | | Time (s) | 176.7 | 358.6 | 1169.4 | 1325.7 | 213.3 |

Two metrics were used to evaluate the performance of the methods:

- **Skeleton F1 Score (SKF1):** The F1 score between the skeleton of the predicted graph and the skeleton of the true graph, which ranges from 0 to 1, where 1 indicates perfect accuracy.

- **Complete PDAG SHD (PSHD):** For the true DAG $\mathcal{G}$ and predicted DAG $\hat{\mathcal{G}}$, PSHD calculates the complete PDAGs of $[\mathcal{G}]$ and $[\hat{\mathcal{G}}]$ and evaluates the number of changes (edge addition, removal, and type change) needed to obtain one PDAG from the other, normalized by the number of nodes. This metric shows the distance between the Markov equivalence classes of the predicted graph and the true graph.

We carried out our experiments in four settings:

- **Low-Dimensional Data:** We evaluated the performance of QWO and other methods on small real-world graphs, namely ASIA [LS88], CANCER [KN10], SACHS [SPP+05], and SURVEY [SD21], as well as $ER2$ graphs with 5 nodes. The number of data samples for this part is set to 500. The results of different methods using both search strategies are presented in Table 2. As shown in the table, QWO demonstrates comparable accuracy to the other approaches for different graph structures while being significantly faster.

- **High-Dimensional Data:** In this setting, we assessed the performance of QWO on larger graphs with 10,000 data samples. Due to the exceedingly long runtime of BDeu and CV General, which are orders of magnitude slower in this setting, they were not included in this experiment. Instead, we compared QWO with the BIC score using both GRaSP and Hill-Climbing search methods. As illustrated in Figure 1, while maintaining high accuracy, QWO demonstrates a significant speedup over BIC for both search methods.

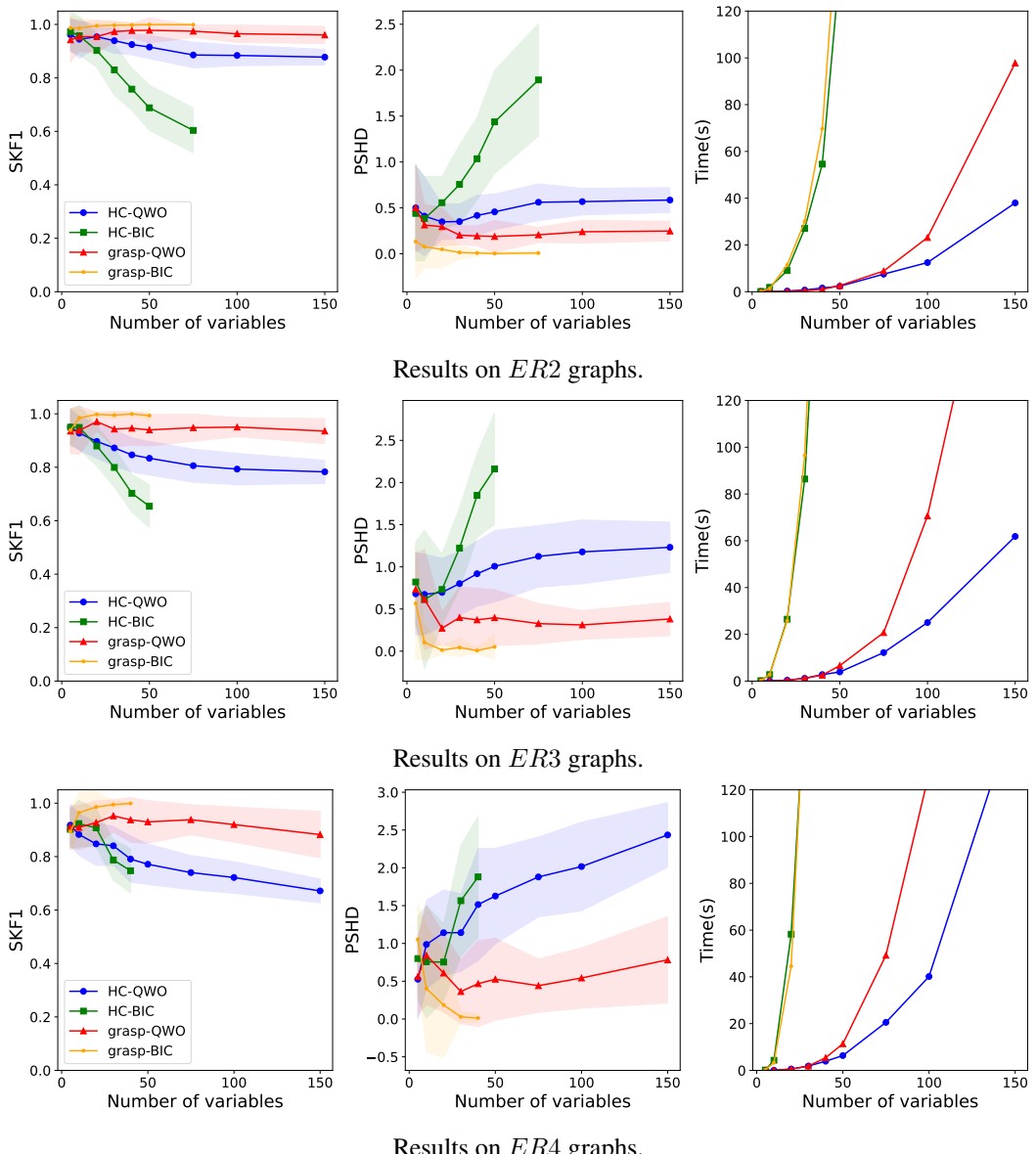

Results on $ER2$ graphs.

Results on $ER3$ graphs.

Results on $ER4$ graphs.

Figure 1: Results of performing QWO and BIC on both search methods GRaSP and HC on data generated by LiGAMs for different Erdos-Renyi graphs ($ER2$, $ER3$, $ER4$).

- **Non-Gaussian Noise Experiments:** To test the robustness of our method, we conducted experiments where the main assumption of Gaussian noise in the data-generating process was violated. We evaluated QWO on linear models with non-Gaussian noise distributions (specifically exponential and Gumbel distributions). The results of these experiments appear in Appendix B. Although QWO is designed for linear models with Gaussian noise, these experiments show that QWO achieves almost similar accuracy to LiGAMs on models with exponential and Gumbel noise distributions.

- **Oracle Inverse Covariance Experiments:** In practice, errors in learning the graph may arise from two sources: the error in calculating the inverse of the covariance matrix and the error in the optimization problem. To study the effect of each error separately, we designed an experiment to eliminate the first source of error by providing the correct inverse covariance matrix (similar to the approach in [DUF+23]). We then compared the performance of the algorithms under this condition. The results of these experiments appear in Appendix B. The plots show that the accuracy

of our method is now significantly high, indicating that the main source of error lies in the initial step of estimating the covariance matrix.

# 6 Conclusion, Limitations, and Future Work

In this paper, we proposed QWO, a module to accelerate permutation-based causal discovery in LiGAMs. Our method reduces the computational complexity of constructing and updating the graph $\mathcal{G}^\pi$ for a given permutation by $O(n^2)$, resulting in a significant speed-up compared to the state-of-the-art BIC-based method, as demonstrated both theoretically and through extensive experiments. Furthermore, QWO seamlessly integrates into existing search strategies, enhancing their scalability without compromising accuracy.

While our method offers substantial improvements, it is important to acknowledge its limitations. The primary assumptions underpinning QWO are that the underlying causal model is a LiGAM—that is, it assumes linear relationships among variables, additive Gaussian noise, and an acyclic causal graph structure. These assumptions, though common in many applications, restrict the applicability of our method to scenarios where these conditions hold. In real-world datasets, relationships may be nonlinear, noise may not be Gaussian, or the causal graph may contain cycles due to feedback loops or reciprocal relationships among variables. Notably, our experiments indicate that QWO maintains competitive performance even when the Gaussian noise assumption is violated, achieving similar accuracy on models with non-Gaussian noise distributions such as exponential and Gumbel (see Appendix B). However, the acyclicity assumption can be particularly limiting in domains where feedback mechanisms are inherent, such as economics, biology, and control systems, where causal graphs are cyclic and methods designed for acyclic graphs are not directly applicable.

Despite this limitation, our work opens avenues for extending permutation-based causal discovery to more general settings. Notably, the set $\mathcal{B}(\mathbf{X})$ is defined over all LiGMs, encompassing both cyclic and acyclic models. Therefore, our characterization of $\mathcal{B}(\mathbf{X})$ in Theorem 4.4 holds for LiGMs as well. Combining this result with our reformulation of causal discovery in LiGAMs in (9), we can generalize the formulation to LiGMs as follows:

$$\underset{Q:QQ^T \in \mathcal{D}_n}{\arg\min} \quad \|I - QW\|_0. \tag{13}$$

This suggests that causal discovery in LiGMs can be approached by finding an orthogonal matrix $Q$ that sparsifies $I - QW$. Some recent works have explored causal discovery in cyclic models using orthogonal transformations similar to ours [GYKZ20]. However, solving the optimization problem in Equation (13) is challenging due to its non-convexity and the orthogonality constraint on $Q$. Developing efficient algorithms to solve this problem is a promising direction for future work.

## Acknowledgments

This research was in part supported by the Swiss National Science Foundation under NCCR Automation, grant agreement 51NF40_180545 and Swiss SNF project 200021_204355 /1.

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

# Appendix

The appendix is organized as follows:

- In Appendix A, we review some of the existing search methods for permutation-based causal discovery.
- In Appendix B, we provide additional experimental results and discuss implementation details.
- In Appendix C, we present formal proofs for the claims made in the main text.

## A  Existing Search Methods

There is a vast literature on permutation-based methods that propose different search strategies in the space of permutations [LAR22, TBA06, SWU21, BSSU20b, TK05]. In this section, we describe two widely used methods: GRaSP and a Hill-Climbing-based method.

### A.1  GRaSP

This method was first introduced in [LAR22]. To present GRaSP, we need two definitions:

**Definition A.1** (Tuck). *Consider any permutation $\pi \in \Pi(n)$ and any $i, j \in [n]$, where $i$ precedes $j$ in $\pi$. Write $\pi$ as $\langle \delta_1, i, \delta_2, j, \delta_3 \rangle$, where each $\delta_i$ is a subsequence of $\pi$. Let $\gamma$ and $\gamma^c$ be the subsequences $\langle t \in \delta_2 : t \in Anc_{\mathcal{G}}(j) \rangle$ and $\langle t \notin \delta_2 : t \in Anc_{\mathcal{G}}(j) \rangle$, respectively. Then define:*

$$tuck(\pi, i, j) = \begin{cases} \langle \delta_1, \gamma, j, i, \gamma^c, \delta_3 \rangle & \text{if } i \in Anc_{\mathcal{G}_\pi}(j) \\ \pi & \text{otherwise} \end{cases}$$

**Definition A.2** (Covered Edge). *In a DAG $\mathcal{G}$, a directed edge $i \to j$ between two nodes is called covered if $Pa_{\mathcal{G}}(i) = Pa_{\mathcal{G}}(j) \backslash \{i\}$.*

[LAR22] proved that starting from any arbitrary permutation $\pi$, there always exists a sequence of permutations $\langle \pi, \pi_1, \pi_2, \cdots, \pi_n \rangle$ such that (i) $\mathcal{G}^{\pi_n} \in [\mathcal{G}]$, (ii) for each $i$, we can reach $\pi_{i+1}$ from $\pi_i$ by a tuck operation, and (iii) $E(\mathcal{G}^{\pi_{i+1}}) \subseteq E(\mathcal{G}^{\pi_i})$ (see Theorem 4.5 in [LAR22]). Subsequently, they propose a greedy algorithm by applying the DFS algorithm on the following graph: A graph with $n!$ vertices, one for each permutation in $\Pi(n)$, and draw a directed edge from node $\pi_1$ to $\pi_2$ if it is possible to find a covered edge $i \to j$ in $\mathcal{G}^{\pi_1}$ and $\pi_2$ is obtained by performing a tuck on the pair $(i, j)$ in $\pi_1$. Finally, they show that under the faithfulness assumption, GRaSP is sound.

### A.2  Hill-Climbing

There are various search methods based on the Hill-Climbing idea, which involve searching over the space of permutations through local changes [TBA06, SG06]. A common way of updating the permutation among these methods is as follows: At each step on a permutation $\pi$, among all pairs $(i, j)$ with $|i - j| \leq k$ (where $k$ is a constant parameter), one pair is chosen randomly. The score is then calculated for the new permutation $\pi_{\text{new}}$ obtained by swapping $\pi(i)$ and $\pi(j)$. If the number of edges in $\mathcal{G}^{\pi_{new}}$ is less than $\mathcal{G}^{\pi}$, the process continues from $\pi_{\text{new}}$. The process stops when no better permutation is found.

## B  Additional Experiments

In this section, we provide implementation details and additional experimental results for the oracle inverse covariance and non-Gaussian noise settings.

### B.1  Implementation Details

We used the implementations provided in the causal-learn library [ZHC$^+$24] for BIC, BDeu, and CV General methods. There are three different versions of GRaSP introduced in [LAR22]. For our experiments, we used GRaSP$_0$. The depth of the DFS algorithm for GRaSP was set to 3, and the value of $k$ (the maximum distance of swapped indices) in the HC algorithm was set to 5.

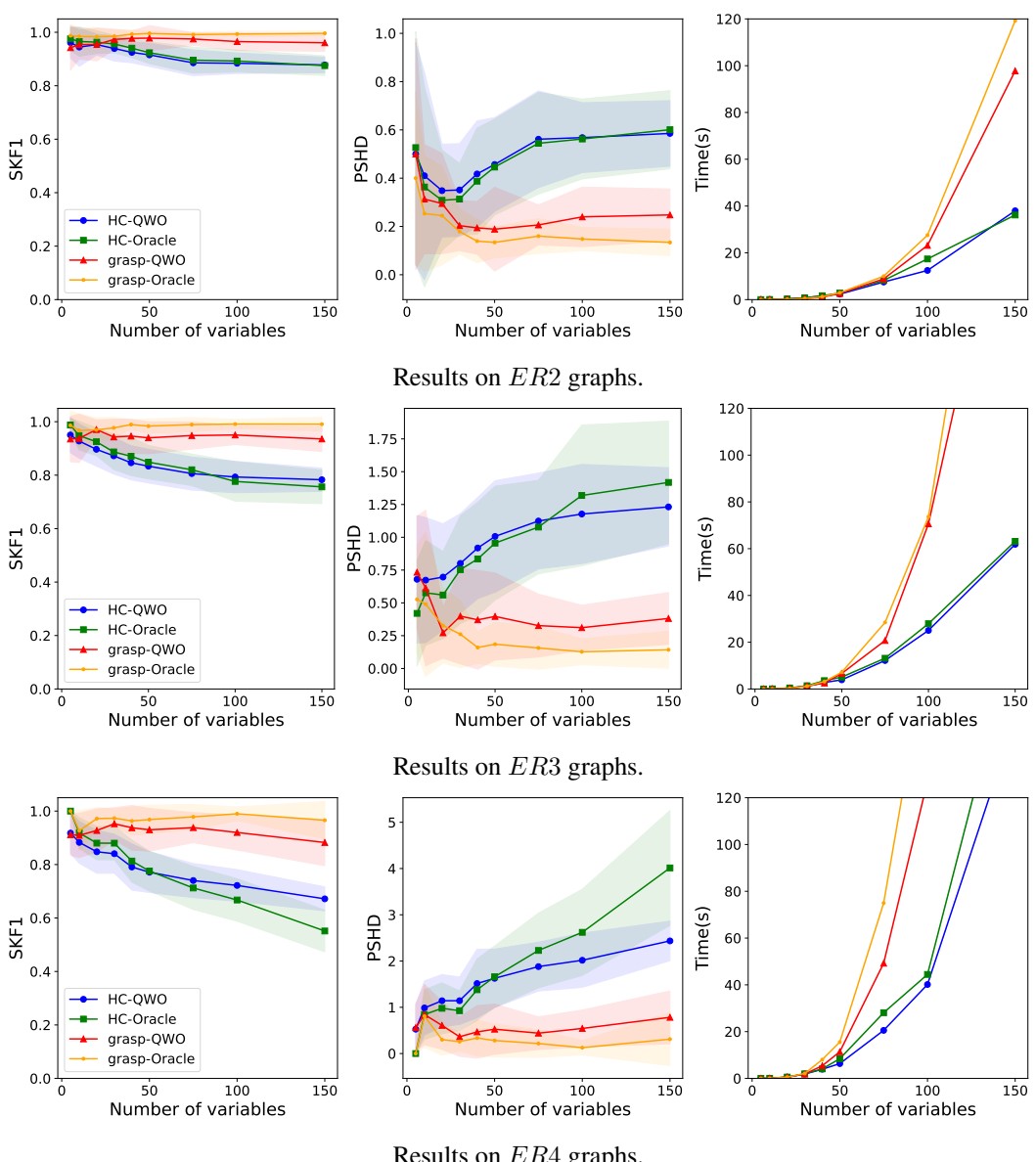

Results on $ER2$ graphs.

Results on $ER3$ graphs.

Results on $ER4$ graphs.

Figure 2: Results for our original method compared to having the true inverse covariance matrix.

For the initial permutation of the search methods, we considered the initial permutation based on the size of Markov boundaries[2] of the variables, which in the case of linear Gaussian models are equal to the number of non-zero elements in each row of the inverse covariance matrix [KF09].

To generate the data matrix $D$ using a linear model $(B^*, \Sigma^*)$, we sampled the entries of $B^*$ uniformly from $[-2, -0.5] \cup [0.5, 2]$ and the noise variances uniformly from $[1, 2]$ for all of the noise distributions i.e., Gaussian, Exponential, and Gumbel.

## B.2 Oracle Inverse Covariance

Figure 2 shows the comparison between our original method, which calculates the inverse covariance from data, and using the oracle inverse covariance matrix. The plots demonstrate that the accuracy of our method is significantly high when using the oracle inverse covariance, indicating that the primary source of error resides in the initial step of estimating the covariance matrix.

---

[2]For the definition of Markov boundary, see [Pea09].

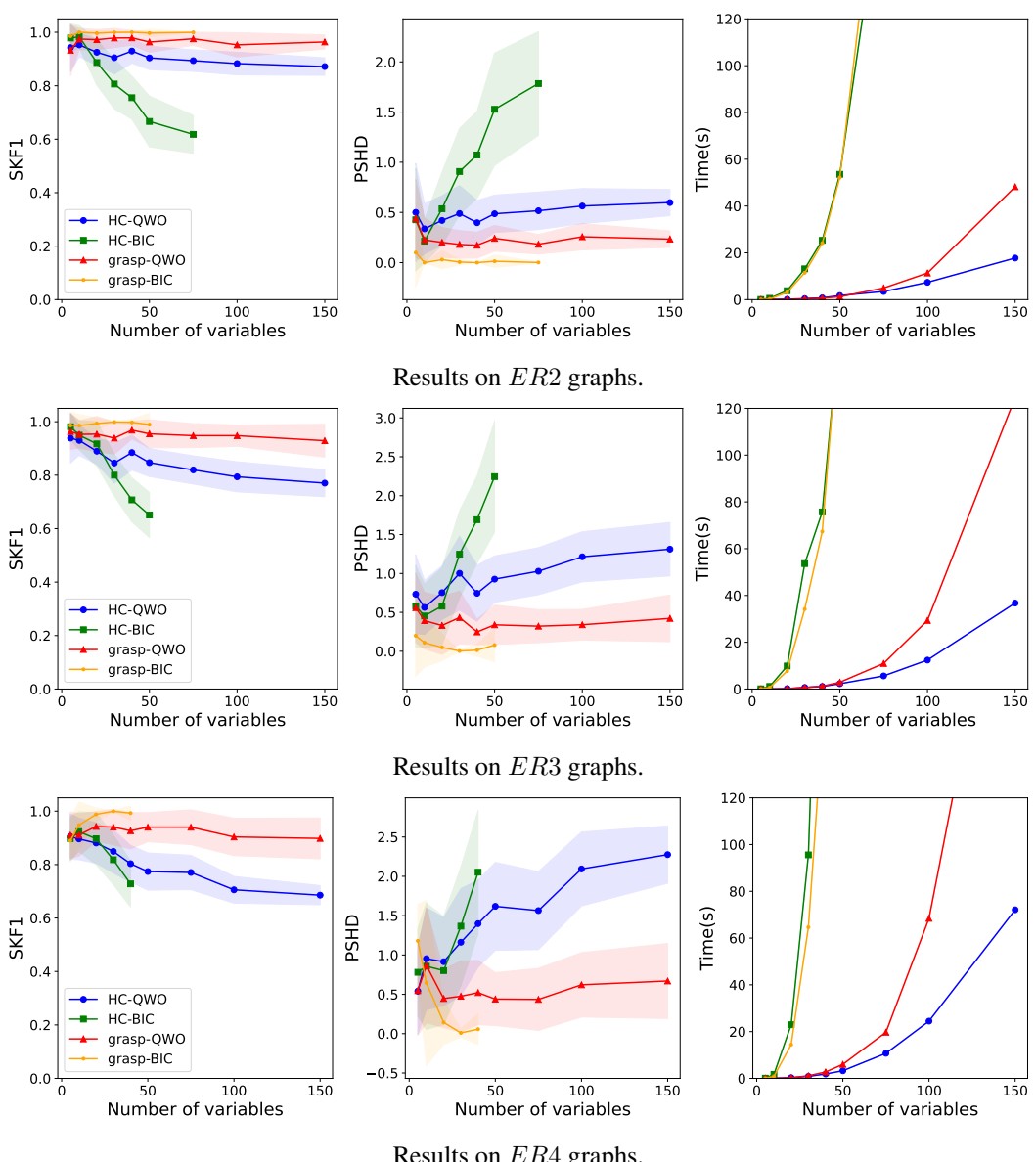

Results on $ER2$ graphs.

Results on $ER3$ graphs.

Results on $ER4$ graphs.

Figure 3: Results for our method compared to BIC on linear models with exponential noise.

### B.3 Non-Gaussian Noise

Herein, we present the results of running QWO and the BIC on two linear **non-Gaussian** models: Exponential and Gumbel noise. Figures 3 and 4 show the time complexity and accuracy in terms of two metrics for both methods on both search strategies. Although QWO is designed for linear models with Gaussian noise, these experiments show that QWO achieves almost similar accuracy to LiGAMs on models with exponential and Gumbel noise distributions.

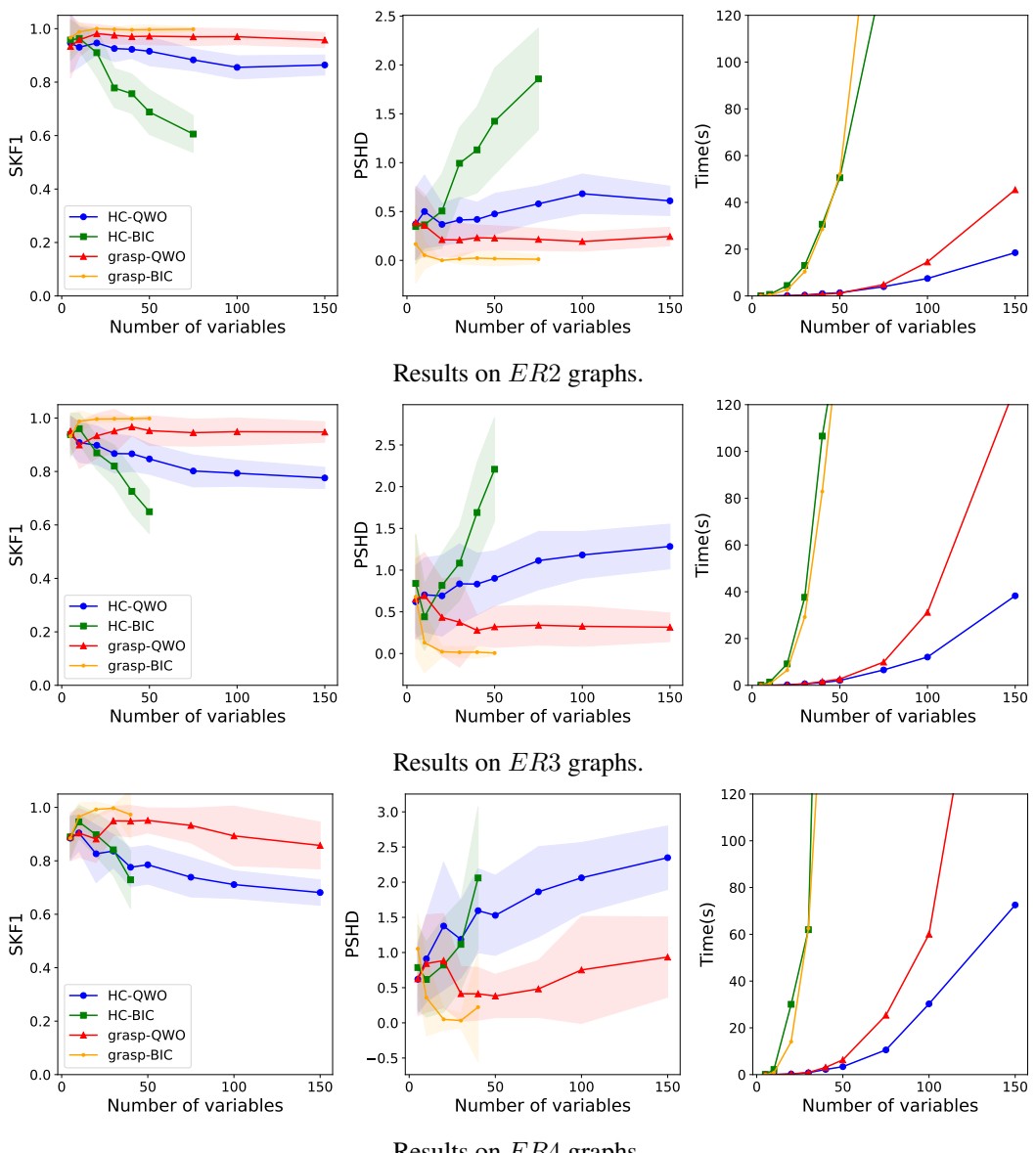

Results on $ER2$ graphs.

Results on $ER3$ graphs.

Results on $ER4$ graphs.

Figure 4: Results for our method compared to BIC on linear models with Gumbel noise.

## C Formal Proofs

**Theorem 4.4** (Characterizing $\mathcal{B}(\mathbf{X})$). *For any $B \in \mathcal{B}(\mathbf{X})$, there exists a unique orthogonal matrix $Q$ such that $B = I - QW$, and vice versa. That is,*

$$\mathcal{B}(\mathbf{X}) = \{I - QW \,|\, QQ^T \in \mathcal{D}_n\}. \tag{11}$$

*Proof.* Recall that

$$\mathcal{B}(\mathbf{X}) = \{B | \exists \Sigma \in \mathcal{D}_n: \quad \mathrm{Cov}(\mathbf{X}) = (I - B)^{-1}\Sigma(I - B)^{-T}\},$$

$\mathrm{Cov}(\mathbf{X}) = USU^T$, and $W = US^{-\frac{1}{2}}U^T$. Therefore, we have

$$W^{-1} = US^{\frac{1}{2}}U^T. \tag{14}$$

First, suppose $B \in \mathcal{B}(\mathbf{X})$, i.e., $\exists \Sigma \in \mathcal{D}_n: \mathrm{Cov}(\mathbf{X}) = (I - B)^{-1}\Sigma(I - B)^{-T}$. We need to show that there exists an orthogonal matrix $Q$ such that $B = I - QW$, or equivalently, $QW = I - B$. Since $W$ is invertible, that is equivalent to showing that $A := (I - B)W^{-1}$ is orthogonal. We have

$$AA^T = \left((I - B)W^{-1}\right)\left((I - B)W^{-1}\right)^T = (I - B)W^{-1}W^{-T}(I - B)^T. \tag{15}$$

Furthermore, Equation (14) implies that

$$W^{-1}W^{-T} = US^{\frac{1}{2}}U^T US^{\frac{1}{2}}U^T = USU^T = \mathrm{Cov}(\mathbf{X}) = (I - B)^{-1}\Sigma(I - B)^{-T}. \tag{16}$$

By substituting the expression for $W^{-1}W^{-T}$ from Equation (16) into Equation (15), we have

$$AA^T = (I - B)(I - B)^{-1}\Sigma(I - B)^{-T}(I - B)^T = \Sigma \in \mathcal{D}_n.$$

Hence, $A$ is orthogonal. The uniqueness of $Q$ for a given $B$ follows from the equation $Q = (I - B)W^{-1}$.

Now suppose $B = I - QW$, where $QQ^T \in \mathcal{D}_n$. We need to show that $B \in \mathcal{B}(\mathbf{X})$. To this end, it suffices to show that $(I - B)\mathrm{Cov}(\mathbf{X})(I - B)^T \in \mathcal{D}_n$. Since $I - B = QW$ and $\mathrm{Cov}(\mathbf{X}) = W^{-1}W^{-T}$, we have

$$(I - B)\mathrm{Cov}(\mathbf{X})(I - B)^T = QW\mathrm{Cov}(\mathbf{X})(QW)^T = QWW^{-1}W^{-T}W^TQ^T = QQ^T \in \mathcal{D}_n.$$

This completes the proof.

$\square$

**Theorem 4.5** (Soundness of Algorithm 1). *Under Assumption 1 and given the correct whitening matrix $W$ as input, matrix $Q$, the output of Algorithm 1 is the unique solution to* (12). *Consequently, the returned graph corresponds to the true $\mathcal{G}^\pi$ defined in Definition 3.1.*

*Proof.* To establish this theorem, we first demonstrate the existence of a unique matrix $Q$ that satisfies both constraints: $\mathrm{diag}(P_\pi QW P_\pi^T) = I$ and $P_\pi QW P_\pi^T$ is upper triangular. Subsequently, we prove that the output of Algorithm 1 is equal to this matrix.

Suppose that $Q$ satisfies both conditions. We proceed by induction on $i$ to show that the $i$-th row of the matrix $P_\pi Q$, denoted by $q_{\pi(i)}^T$, is unique. The constraint that $P_\pi QW P_\pi^T$ is upper triangular implies that $\forall i < j : q_{\pi(i)} \perp w_{\pi(j)}$.

Induction base, i.e., when $i = 1$: The vector $q_{\pi(1)}$ is orthogonal to all of the vectors in $\{w_{\pi(2)}, w_{\pi(3)}, \ldots, w_{\pi(n)}\}$ which determines the unique direction of $q_{\pi(1)}$. Moreover, $\mathrm{diag}(P_\pi QW P_\pi^T) = I$ implies $\langle q_{\pi(1)}, w_{\pi(1)} \rangle = 1$. These two conditions show that $q_{\pi(1)}$ is unique.

Induction step: suppose $\{q_{\pi(1)}, q_{\pi(2)}, \ldots, q_{\pi(i-1)}\}$ are unique and we want to show that $q_{\pi(i)}$ is unique. Note that $q_{\pi(i)}$ is orthogonal to vectors in both sets $\{q_{\pi(1)}, q_{\pi(2)}, \ldots, q_{\pi(i-1)}\}$ and $\{w_{\pi(i+1)}, w_{\pi(i+2)}, \ldots, w_{\pi(n)}\}$, and vectors in the second set are orthogonal to the first set and also each other. This shows that all of the vectors in both sets are linearly independent, and the direction of $q_{\pi(i)}$ is unique. Therefore, $\mathrm{diag}(P_\pi QW P_\pi^T) = I$ implies the uniqueness of $q_{\pi(i)}$.

Now, to prove the theorem, it suffices to show that the matrix $Q$, the output of Algorithm 1 satisfies $\mathrm{diag}(P_\pi QW P_\pi^T) = I$ and $P_\pi QW P_\pi^T$ is upper-triangular. For the first one, note that in

line 6 of the algorithm, vectors $q_{\pi(i)}$ are normalized such that $\langle q_{\pi(i)}, w_{\pi(i)} \rangle = 1$ which implies $\text{diag}(P_\pi Q W P_\pi^T) = I$. For the second condition, note that for each $1 \le i \le n$, vector $q_{\pi(i)}$ is the normalized residual of the projection of $w_{\pi(i)}$ into the space of $\{w_{\pi(i+1)}, \ldots, w_{\pi(n)}\}$, which shows the orthogonality of $q_{\pi(i)}$ to $w_{\pi(j)}$ with $j > i$. This property implies $P_\pi Q W P_\pi^T$ is upper-triangular. $\square$

**Theorem 4.2.** *Under Assumption 1, for any permutation $\pi \in \Pi([n])$, there exists a unique $B \in \mathcal{B}(\mathbf{X})$ such that $G(B)$ is compatible with $\pi$. Furthermore, for this $B$, $G(B) = \mathcal{G}^\pi$.*

*Proof.* In Theorem 4.5, we showed that (12) has a unique solution and then proved that the output of Algorithm 1 is equal to this solution. To complete the proof, it suffices to prove that graph $G(P_\pi Q W P_\pi^T)$ is equal to $\mathcal{G}^\pi$, where $Q$ is the output of Algorithm 1. To show this, we need to prove for each $i < j$,

$$X_{\pi(i)} \perp\!\!\!\perp X_{\pi(j)} | X_{\{\pi(1),\pi(2),\ldots,\pi(j-1)\}\setminus\{\pi(i)\}} \iff \langle q_{\pi(j)}, w_{\pi(i)} \rangle = 0. \tag{17}$$

To prove (17), we first need a few notations. For a matrix $A$, $A[i:j]$ denotes the matrix consisting of **rows** $\{i, i+1, \ldots, j\}$ of matrix $A$. Similarly, $A[i]$ denotes the $i$−th **row** of $A$. For a vector $v$ and a set of vectors $\mathbf{u} = \{u_1, u_2, \ldots, u_k\}$, $proj_\mathbf{u}(v)$ and $res_\mathbf{u}(v)$ denote the projection of vector $v$ on the span of $\mathbf{u}$, and the residual of this projection, respectively, i.e., $res_\mathbf{u}(v) = v - proj_\mathbf{u}(v)$. For a matrix $A$, $res_A(v)$ denotes the projection of vector $v$ on the space spanned by the rows of $A$.

Now, let us fix $1 \le i < j \le n$. In the rest of the proof, we will prove Equation (17) for $i, j$. Recall that $Cov(X) = USU^T$ denotes the SVD decomposition of the covariance matrix of $\mathbf{X}$. Define

$$A = U[1:j], \quad B = U[j+1:n], \quad C = W[1:j], \quad D = W[j+1:n].$$

Note that $AA^T = I_j$ and $BB^T = I_{n-j}$, where $I_k$ denotes the $k \times k$ identity matrix. Furthermore,

$$C = AS^{-\frac{1}{2}}U^T, \quad D = BS^{-\frac{1}{2}}U^T. \tag{18}$$

We further define $E$ to be the $j \times n$ matrix constructed as follows:

$$E[k] = res_D(C[k]^T)^T, \quad \forall 1 \le k \le j. \tag{19}$$

To prove (17), we present the following two lemmas.

**Lemma C.1.** *(Block Matrix Inversion Lemma [Ber09]) Suppose $T$ is an invertible matrix, which is in the following form.*

$$T = \begin{bmatrix} T_{11} & T_{12} \\ T_{21} & T_{22} \end{bmatrix}$$

*In this case, $T^{-1}$ can be computed using the Schur complement of $T_{11}$ as follows:*

$$T^{-1} = \begin{bmatrix} M & -T_{11}^{-1}T_{12}N^{-1} \\ -N^{-1}T_{21}T_{11}^{-1} & N^{-1} \end{bmatrix},$$

*where $N = T_{22} - T_{21}T_{11}^{-1}T_{12}$ is the Schur complement of $T_{11}$ in $T$ and is assumed to be invertible, and $M = (T_{11} - T_{12}T_{22}^{-1}T_{21})^{-1}$.*

**Lemma C.2.** *For matrix $E$ defined in (19), the following holds:*

$$EE^T = Cov(\mathbf{X}_{[j]})^{-1} \tag{20}$$

*Recall that $\mathbf{X}_{[j]} = [X_1, \ldots, X_j]^T$.*

*Proof.* We will prove that $(EE^T)^{-1} = Cov(\mathbf{X}_{[j]})$. We define

$$P = D^T(DD^T)^{-1}D,$$

which is the projection matrix that projects any vector to the space of rows of $D$. Therefore, we have

$$E = C - (PC^T)^T = C(I - P^T). \tag{21}$$

Next, we calculate $DD^T$, $P$, and $EE^T$ in terms of $A, B$, and $S$. First, (18) implies the following.

$$DD^T = BS^{-\frac{1}{2}}U^T(BS^{-\frac{1}{2}}U^T)^T = BS^{-\frac{1}{2}}U^TUS^{-\frac{1}{2}}B^T = BS^{-1}B^T. \tag{22}$$

Using (22), we have

$$P = D^T(DD^T)^{-1}D = US^{-\frac{1}{2}}B^T(BS^{-1}B^T)^{-1}BS^{-\frac{1}{2}}U^T. \tag{23}$$

Using (21), we have

$$EE^T = C(I - P^T)(C(I - P^T))^T = C(I - P)(I - P)^TC^T = C(I - P)C^T$$
$$= CC^T - CPC^T. \tag{24}$$

Note that in (24), we used $(I - P)(I - P)^T = I - P$, which holds true because $P$ is a projection matrix. To further refine (24), we apply (23) to calculate $CC^T$ and $CPC^T$ in the following.

$$CC^T = AS^{-\frac{1}{2}}U^T(AS^{-\frac{1}{2}}U^T)^T = AS^{-\frac{1}{2}}U^TUS^{-\frac{1}{2}}A^T = AS^{-1}A^T$$
$$CPC^T = AS^{-\frac{1}{2}}U^T\left(US^{-\frac{1}{2}}B^T(BS^{-1}B^T)^{-1}BS^{-\frac{1}{2}}U^T\right)(AS^{-\frac{1}{2}}U^T)^T$$
$$= AS^{-1}B^T(BS^{-1}B^T)^{-1}BS^{-1}A^T$$

Applying the last equations in (24), we have

$$EE^T = AS^{-1}A^T - AS^{-1}B^T(BS^{-1}B^T)^{-1}BS^{-1}A^T. \tag{25}$$

Next, we present $\mathrm{Cov}(\mathbf{X})^{-1}$ in terms of $A, B, S$.

$$\mathrm{Cov}(\mathbf{X})^{-1} = US^{-1}U^T = \begin{bmatrix} A \\ B \end{bmatrix} S^{-1} \begin{bmatrix} A & B \end{bmatrix} = \begin{bmatrix} AS^{-1}A^T & AS^{-1}B^T \\ BS^{-1}A^T & BS^{-1}B^T \end{bmatrix}. \tag{26}$$

Applying Lemma C.1 to $\mathrm{Cov}(\mathbf{X})^{-1}$ with (26), we have

$$\mathrm{Cov}(\mathbf{X}) = \begin{bmatrix} M & -(AS^{-1}A^T)^{-1}AS^{-1}B^TN^{-1} \\ -N^{-1}BS^{-1}A^T(AS^{-1}A^T)^{-1} & N^{-1} \end{bmatrix},$$

where

$$M = (AS^{-1}A^T - AS^{-1}B^T(BS^{-1}B^T)^{-1}BS^{-1}A^T)^{-1}, \tag{27}$$

and $N$ is a matrix that we do not need to calculate. Note that $M = \mathrm{Cov}(\mathbf{X}_{[j]})$ since $M$ is an $j \times j$ matrix. Furthermore, (27) and (25) imply that $M = (EE^T)^{-1}$. Therefore, $(EE^T)^{-1} = \mathrm{Cov}(\mathbf{X}_{[j]})$, which concludes the proof. □

Now to prove the theorem we use a classic result on the linear Gaussian data. Consider $\Theta$ is the inverse of the covariance matrix for some variables with joint Gaussian distribution, then two variables $X_i$ and $X_j$ are independent given all the other variables if and only if $\Theta_{i,j} = 0$ [KF09]. Considering this result, to show (17), it is sufficient to show the following:

$$\langle q_{\pi(j)}, w_{\pi(i)} \rangle = 0 \iff (\mathrm{Cov}(\mathbf{X}_{[j]})^{-1})_{i,j} = 0$$

To show the above equation, let $D^\pi$ denotes the set $\{w_{\pi(j+1)}, w_{\pi(j+2)}, \ldots, w_{\pi(n)}\}$, then $q_{\pi(j)} = res_{D^\pi}(w_{\pi(j)})$. Thus, $q_{\pi(j)}$ is orthogonal to the span of the vectors in $D^\pi$, then we have

$$\langle w_{\pi(i)}, q_{\pi(j)} \rangle = \langle w_{\pi(i)}, res_{D^\pi}(w_{\pi(j)}) \rangle = \langle res_{D^\pi}(w_{\pi(i)}) + proj_{D^\pi}(w_{\pi(i)}), res_{D^\pi}(w_{\pi(j)}) \rangle$$
$$= \langle res_{D^\pi}(w_{\pi(i)}), res_{D^\pi}(w_{\pi(j)}) \rangle$$

Based on lemma C.2, the value of $\langle res_{D^\pi}(w_{\pi(i)}), res_{D^\pi}(w_{\pi(j)}) \rangle$ is equal to $(\mathrm{Cov}(\mathbf{X}_{[j]})^{-1})_{i,j}$ and this result completes the proof of the theorem.

□

**Lemma 4.6.** *If the block between the $idx_l$-th and $idx_r$-th positions of $\pi$ is modified, the vectors $q_{\pi(k)}$ for $k < idx_l$ or $k > idx_r$ remain unchanged.*

*Proof.* Let $\pi'$ be the new permutation obtained after modifying the permutation $\pi$. First let $k > \text{idx}_r$. In this case, for each $j > \text{idx}_r$, $w_{\pi(j)} = w_{\pi'(j)}$, thus, $r_j$ remains the same for both $\pi$ and $\pi'$. Therefore, $q_{\pi(k)} = q_{\pi'(k)}$.

Now let $k < \text{idx}_l$. As we discussed in the main text, due to the construction of matrix $Q$ in Algorithm 1, for any $1 \leq i \leq n$, the span of vectors $\{q_{\pi(i)}, q_{\pi(i+1)}, \ldots, q_{\pi(n)}\}$ is equal to the span of vectors $\{w_{\pi(i)}, w_{\pi(i+1)}, \ldots, w_{\pi(n)}\}$. Furthermore, since we just modified the block between the $\text{idx}_l$-th and $\text{idx}_r$-th positions of $\pi$ to obtain $\pi'$, the two sets $\{\pi(k+1), \pi(k+2), \ldots, \pi(n)\}$ and $\{\pi'(k+1), \pi'(k+2), \ldots, \pi'(n)\}$ are equal. Therefore, the sets $\{w_{\pi(k+1)}, w_{\pi(k+2)}, \ldots, w_{\pi(n)}\}$ and $\{w_{\pi'(k+1)}, w_{\pi'(k+2)}, \ldots, w_{\pi'(n)}\}$ are also equal. Hence, the residual of vector $w_{\pi(k)}$ on these two sets is equal, which implies that $q_{\pi(k)} = q_{\pi'(k)}$. This completes the proof. $\square$

**Theorem 4.7** (Time complexity of Algorithm 1). *QWO algorithm as implemented in Algorithm 1 has the following time complexities:*

- *$O(n^3)$ for initially computing $\mathcal{G}^\pi$ without optional arguments.*

- *$O(n^2 d)$ when called with optional arguments to update $\mathcal{G}^\pi$, where $d = idx_r - idx_l$.*

*Proof.* In Algorithm 1, the steps outside the for loop include initialization of variables and computing $G(I - QW)$ are executed in $O(n^2)$. Inside the for loop, the time complexity for calculating each $r_i$ is $O(n(n-i))$ because each term in the summation is done in $O(n)$. As $i$ iterates from 1 to $n$, this complexity is $O(n^2)$ and for the entire loop it takes $O((\text{idx}_r - \text{idx}_l)n^2)$. Therefore, the time complexity for the initial step is $O(n^3)$, and for the update steps is $O(n^2 d)$. $\square$

