# OpenReview forum: "QWO: Speeding Up Permutation-Based Causal Discovery in LiGAMs"
_NeurIPS.cc/2024/Conference — NeurIPS 2024 poster_

### Official Review · Reviewer_Znq2 · 2024-07-10

**Soundness:** 2
**Presentation:** 2
**Contribution:** 3
**Rating:** 5
**Confidence:** 3

**Summary:**

This paper focuses on permutation-based methods for causal discovery in Linear Gaussian Acyclic Models (LiGAMs). A new method called QWO is proposed to improve the efficiency of computing a causal graph given a permutation. Compared with baselines, QWO achieves superior performance.

**Strengths:**

1.	According to the theoretical analysis, the proposed method is guaranteed to learn the true graph for a given permutation when there are sufficient samples.
2.	The computational complexity is quadratic, much lower than other methods, such as the BIC-based method.
3.	QWO can be applied to some other methods as an accerating module.
4.	The experiments are sufficient to support their claims.

**Weaknesses:**

1.	The writing is poor. The format for definitions is not consistent throughout the paper. Some definitions are directly given in the content body (Lines 64-72), while some others are put in a ‘Definition’ framework (Definition 2.1, 3.1, 3.2). Some contents are repetitive. Line 89 and Line 94 both mention that “faithfulness holds”.
2.	Some definitions are not properly expressed. In Definition 3.3., how come “there is an edge from $X_{\pi(i)}$ to $X_{\pi(j)}$ if and only if $X_{\pi(i)}$ and $X_{\pi(j)}$ are conditionally independent”?
3.	Assumptions are not clearly demonstrated. Despite faithfulness, Markov assumption is also required for inferring causal graphs.

**Questions:**

(See above)

**Limitations:**

Yes. The assumptions are restricted, such as the linear Gaussian model and faithfulness assumption.

---

> ### Author Rebuttal · Authors · 2024-08-07
>
> We thank the reviewer for acknowledging our method's theoretical guarantees, efficient computational complexity, and of our experimental results.
>
> ---
> > The format for definitions is not consistent throughout the paper. Some definitions are directly given in the content body (Lines 64-72), while some others are put in a ‘Definition’ framework (Definition 2.1, 3.1, 3.2).
>
> We used the ‘Definition’ framework for the following cases:
>
> - Definition 2.1 $G(B)$
>
> - Definition 3.1 $[\mathcal{G}]$
>
> - Definition 3.2 $\mathcal{B}(X)$
>
> - Definition 3.3 $\mathcal{G}^{\pi}$
>
> - Definition 4.2 Whitening matrix $W$
>
> These notations are either new and crucial for our presentation, or there is no consensus for them in the literature. On the other hand, the notations and definitions provided in lines 64-72 (which is the notations section) are commonly used in various fields, and we did not see the necessity of assigning a separate box for them.
>
> ---
> > Line 89 and Line 94 both mention that “faithfulness holds”.
>
> We will edit line 89.
>
> ---
> >  In Definition 3.3., how come “there is an edge from $X_{\pi(i)}$ to $X_{\pi(j)}$ if and only if $X_{\pi(i)}$ and $X_{\pi(j)}$ are conditionally independent”?
>
> We thank the reviewer for pointing out this typo. The independency ($\perp \mathrel{\mkern-9mu} \perp$) in Equation (5) should be changed to dependency.
>
> ---
> > Despite faithfulness, Markov assumption is also required for inferring causal graphs.
>
> When the underlying model is a structural equation model (SEM), which is a more general setting of our problem, the Markov property holds (please refer to Theorem 1.2.5 in [1]). Therefore, Markov property is not technically an assumption. To avoid any confusion, we will mention this in the revised version.
>
> [1] Pearl, Judea. Causality. Cambridge university press, 2009.

---

> > ### Comment · Reviewer_Znq2 · 2024-08-12
> >
> > Thanks for the responses, and I will maintain my score.

---

### Official Review · Reviewer_rQrH · 2024-07-11

**Soundness:** 4
**Presentation:** 4
**Contribution:** 3
**Rating:** 7
**Confidence:** 3

**Summary:**

The authors present an efficient method for evaluating a score for score-based causal discovery over LiGAMs. Their method uses the whitening matrix $W$, derived from the observed covariance matrix, as a summary statistic. Their method is $\mathcal{O}(n^2)$ faster than the classical *BIC* method, where $n$ is the number of observed variables.

Because $W$ only needs to be calculated once, the authors' score can be evaluated equally efficiently for any number of samples $N$. This presents an advantage against state-of-the-art methods such as *BDeu* and *CV General*, which become difficult for large $N$ and are primarily intended for nonlinear models.

Their score for each topological ordering is the fewest edges a DAG can have while being consistent with at least one LiGAM that produces the observed covariance matrix.

**Strengths:**

The authors present an elegant and intuitive approach with a well-motivated score function. They write with clarity and demonstrate the clear advantage of their method over the state-of-the-art in simulated and real-world experiments.

**Weaknesses:**

The authors' method does not address finite-sample uncertainty and is best thought of as providing a (discrete) point estimate of an underlying population score that is a function of the population whitening matrix $W^*$. It would have been impressive for the authors to handle finite-sample uncertainty in $W^*$ as a part of their approach.

In *BIC*, for example, the likelihood of the data given a graph is explicitly part of the score function. As the number of samples $N$ increases, *BIC* will provide increasingly lower scores to DAGs outside of the Markov equivalence class of the true DAG $G^*$. This does not happen for QWO since $W$ is treated as a population whitening matrix either way.

**Questions:**

Can QWO be extended to incorporate finite-sample uncertainty while retaining (some of) the advantage in the scalability of the method? For instance, could a confidence set over scores be output with the same scalability?

Perhaps not and QWO is intended primarily for the setting with large $N$ where other methods are too computationally expensive?

**Limitations:**

Yes, the required assumptions are stated.

---

> ### Author Rebuttal · Authors · 2024-08-07
>
> We thank the reviewer for their interesting comment regarding incorporating finite-sample uncertainty. We also appreciate the positive feedback on our method's clarity, intuitive design, and superiority in experiments.
>
> ---
> > Can QWO be extended to incorporate finite-sample uncertainty while retaining (some of) the advantage in the scalability of the method? For instance, could a confidence set over scores be output with the same scalability?
>
> Below, we drive an uncertainty analysis for each edge in $\mathcal{G}^{\pi}$.
> In our experiments, our method incorporates a point estimate of $W$ using a finite set of samples. This estimation can indeed be noisy, which is the primary source of error for the final estimation. After constructing $q_1, q_2, ..., q_n$, we then proceed to check the orthogonality of vectors  $w_1, ..., w_n$ and $q_1, ..., q_n$.
> We can show that for $i<j$, the dot product of $w_{\pi(i)}$ and $q_{\pi(j)}$ is an estimation for partial correlation $\rho_{X_{\pi(i)}, X_{\pi(j)} |{X_{\{\pi(1), \pi(2), \ldots, \pi(j-1)\}\backslash\{\pi(i)\}}}}.$
> To check whether this dot product is zero, we then apply a z-test for this partial correlation.
> Using results in [1], we can accordingly derive the following confidence interval:
> $$ P(|\text{Error of estimation}|>\epsilon) \leq \frac{1}{ (\text{NumOfSamples} -j -2) \epsilon^2},
> $$
> where `Error of estimation' is the distance between our estimated dot product and the actual dot product.
>
> [1] Drton, Mathias, and Michael D. Perlman. "Multiple testing and error control in Gaussian graphical model selection." (2007): 430-449.
>
> ---
> > QWO is intended primarily for the setting with large $N$ where other methods are too computationally expensive?
>
> Our method has the advantage of being fast with large sample sizes, but it can also be applied in situations with small sample sizes. It is worth noting that there is extensive literature on computing the inverse covariance matrix with a small number of data points. For example, see [2, 3].
>
> [2] Ravikumar, Pradeep, et al. "High-dimensional covariance estimation by minimizing l1-penalized log-determinant divergence." (2011): 935-980.
>
> [3] Friedman, Jerome, Trevor Hastie, and Robert Tibshirani. "Sparse inverse covariance estimation with the graphical lasso." Biostatistics 9.3 (2008): 432-441.

---

> > ### Comment · Reviewer_rQrH · 2024-08-12
> >
> > Many thanks for the clarifications. I maintain my score 7: Accept.

---

### Official Review · Reviewer_D3Yo · 2024-07-12

**Soundness:** 3
**Presentation:** 4
**Contribution:** 3
**Rating:** 7
**Confidence:** 4

**Summary:**

Authors propose a new causal discovery algorithm on permutation-based methods in the context of Linear Gaussian Acyclic Models (LiGAMs). Specifically, authors focus on the computation complexity of existing solutions and propose a novel QW-Orthogonality (QWO) that improve the efficiency of computing a new graph $\mathcal{G}$ for a given permutation $\pi$. The computational complexity of QWO is $\mathcal{O}(n^2)$, that is, significantly better than BIC-based alternatives.

**Strengths:**

- The outline of the proposed solution is clear: the computational complexity of alternative solutions is a strong limitation for the applicability of causal discovery methods. Recasting the optimization problem is the key of the contribution and it's a original contribution.
- The quality of the paper is relevant: methods are self-contained, they are clear and easy to follow.
- The significance of the proposed solution is evident from the experimental setup, where existing solutions exceed the running time caps.

**Weaknesses:**

- The proposed solution is tested in combination of just two search-based procedures, which is a rather limited evaluation, especially if we observe that the gap between HC-BIC and GRASP-BIC is significant.
- GRASP-BIC usually achieves better results then the proposed solution in random graphs.

**Questions:**

- Why comparing PDAGs instead of CPDAGs when computing evaluation metrics?

**Limitations:**

- Authors do not discuss explicitly the limitations of the proposed method, they refer to the theoretical assumptions instead.
- The assumptions are the usual ones that are present in every causal discovery algorithm.

---

> ### Author Rebuttal · Authors · 2024-08-06
>
> We appreciate the reviewer's comments and are pleased that they found our approach to be a clear and original contribution.
>
> ---
> > Why comparing PDAGs instead of CPDAGs when computing evaluation metrics?
>
> We thank the reviewer for pointing out this issue/typo. In our experiments, we have indeed compared the CPDAGs (complete PDAG achieved after applying Meek rules). We will correct this typo in the revised version.
>
> ---
> > GRASP-BIC usually achieves better results then the proposed solution in random graphs.
>
> We acknowledge the reviewer's observation that GRASP-BIC usually archives slightly higher accuracy in random graphs.
> However, we note that (i) the difference in accuracy is almost negligible, and (ii) the primary goal of our method is to reduce computational complexity, which we have demonstrated both theoretically (by a factor of $O(n^2)$) and empirically. Figure 1 shows that GRASP-BIC was scalable to nearly 50 variables and only in sparse graphs, while our method was easily scalable to over 150 variables, even in denser graphs.

---

> > ### Comment · Reviewer_D3Yo · 2024-08-11
> >
> > Thank you taking your time to reply. I'm satisfied with the explanations provided.

---

### Official Review · Reviewer_9HgY · 2024-07-15

**Soundness:** 3
**Presentation:** 3
**Contribution:** 2
**Rating:** 5
**Confidence:** 4

**Summary:**

This paper considers the problem of speeding up permutation-based causal discovery in linear Gaussian acyclic models. A typical permutation-based causal discovery algorithm includes two components: 1) constructing a DAG permitting a given topological ordering, and 2) a search strategy over the space of permutations. While most existing work focuses on component 2), this work focuses on 1). Specifically, the authors first characterize the adjacency matrices' equivalence class using whitening transformation and orthogonal rotation. Then, with this intuition, an algorithm QW-Orthogonality (QWO) based on Gram-Schmidt algorithm is proposed to construct the DAG under a given ordering. This QWO algorithm has a time complexity of O(n3) for an ordering pi without side information.

**Strengths:**

+ It is novel to study the speeding up of component 2 (constructing a DAG permitting a given topological ordering) when most of the existing work considers the search strategy over permutations.

+ The proposed algorithm can be integrated into various existing permutation-based algorithms and enjoys a lower time complexity in updating steps.

+ The review on existing work regarding permutation-based algorithms and the overall writing is clear.

**Weaknesses:**

+ **Assumptions needed is not spelled out:** What is the exact assumptions needed for this work? Is it normal faithfulness, or sparsest Markovian representation as in [RU18], or something in between as in [LAR22]? Since a primary focus of permutation-based algorithms is to relax assumptions needed, I would suggest authors put assumptions explicitly (and check if they are sufficient, or sufficient and necessary). Since different lemmas/theorems may need different assumptions, it is better to specify assumptions for each of them separately.

+ **The significance of the proposed method may need a better justification:** The authors defined the whole equivalence class $\mathcal{B}$ that involves also cyclic graphs and those even with self-loops. The introduction of orthogonal transformation also reflects this. However, in this work actually only acyclic graphs are considered (as seen in the upper triangular constraints in (13)). In this case, why do the authors take a (seemingly) detour for orthogonal transformation, instead of directly apply Cholesky/LDL transformation on the permuted covariance matrix? The time complexity of this is also O(n3). With some small modification to (constrained) Cholesky decomposition, I guess an updating complexity of O(n2d) can also be achieved.

+ **A more clearer justification for the necessity of speeding up Gπ construction is needed:** For example, in line 138, "to solve (9), ..., Note that the complexity of a brute-force search over all subsets U is exponential." I don't quite get this -- why do we need to try all subsets as parents candidates, given that by definition (5), the existence of each edge can directly be seen without any sense of subset traversing -- though the time complexity of directly using (5) is generally O(n5) (O(n2 for testing each edge) and O(n3) for each Pearson correlation calculation).

+ **Some missing references:** For example, the idea to use orthogonal rotation to characterize the equivalence class is very similar to https://arxiv.org/abs/1910.12993. The similarities and differences should be discussed.

**Questions:**

As in "Weaknesses".

---

> ### Author Rebuttal · Authors · 2024-08-06
>
> We thank the reviewer for their detailed and thoughtful comments.
>
> ---
> > Assumptions needed is not spelled out
>
> We agree with the reviewer that permutation-based algorithms aim to relax necessary assumptions, such as faithfulness. However, for our method, the sparsest Markovian representation assumption in [RU18] is indeed sufficient. We did not mention this in the text to keep it simple and avoid exceeding the page limit. Since we have an extra page for the revised version, we will explicitly mention this lesser version of faithfulness. Additionally, for the sake of completeness, we will refer to the exact assumptions in the main results of the paper.
>
> ---
> > The significance of the proposed method may need a better justification
>
> We acknowledge the reviewer's insights on the matter. While there may be alternative methods based on Cholesky decomposition as the reviewer suggested, our method offers some advantages. For instance, the update step in our method is straightforward and can be easily integrated into existing search methods. Furthermore, we propose a characterization over $\mathcal{B}(X)$ in Theorem 4.3, which is defined for cyclic models as well, paving the way for learning cyclic models. For example, by removing the upper-triangularity constraint in Equation 13, it would be a valid optimization for learning a cyclic graph (though we might need a stronger version of faithfulness to extend this to cyclic models). However, solving the optimization for cyclic models could be a challenging problem, but it is a promising direction for future work.
>
>
> ---
> > A more clearer justification for the necessity of speeding up $\mathcal{G}^{\pi}$ construction is needed
>
> We believe there has been a misunderstanding regarding lines 138-141, which we include here for reference:
>
> > To solve (9), various approaches have been proposed. Note that the complexity of a brute-force search over all subsets $\mathbf{U}$ is exponential. Instead, the state-of-the-art search methods apply the grow-shrink (GS) algorithm [ARSR+23] on the candidate sets $\mathbf{U}$ to find the parent set of each variable. These methods require computing the score function $S$, $O(n^2)$ times.
>
> Herein, we provide a literature review of the existing methods for constructing $\mathcal{G}^{\pi}$. We start by pointing out that brute force is not a feasible method. Therefore, we mention alternative approaches that involve computing the score function $S$ only $O(n^2)$ number of times. We then introduce different score functions for $S$ in the following paragraph. This gives us the computation complexity of the existing work for constructing $\mathcal{G}^{\pi}$ (please refer to Table 1). Finally, we justify our proposed method by the fact that it gains a speed-up of $O(n^2)$ in comparison to the aforementioned class of approaches.
>
> ---
> > Some missing references
>
> We appreciate the reviewer for bringing up that paper. It looks very interesting and relevant to our problem. In the revised version, we will include a brief discussion on the similarities and differences between our work and this paper.

---

### Decision · Program_Chairs · 2024-09-25

**Decision:**

Accept (poster)

**Comment:**

All reviewers agree on the merits of this manuscript, that framing learning a CPDAG for Linear Gaussian models as an optimization problem is interesting and worthwhile by itself, as well as that it may lead to novel approaches to adjacent problems. There is also agreement that the write-up needs improvement, such as clearly stating the assumptions under which guarantees can be given. Similarly, the authors should more carefully explain what can be done for cyclic resp. acyclic models, and support this with empirical evidence. Last, the authors are strongly recommended to clearly discuss the limitations of the method beyond merely stating the assumptions. I would recommend them to also evaluate and discuss how well the method works when the assumptions are not met (as they probably wont hold in practice anyway). All in all, the general sentiment for this paper is positive and hence an accept is in order. Please use the additional page to address the further improve the manuscript, especially the three points above.